**Data Availability Statement:** All relevant data are within the paper.

**Funding:** This research was supported by the Department of Science and Technology, India

# Feasibility and usability of a virtual-reality-based sensorimotor activation apparatus for carpal tunnel syndrome patients

Kishor Lakshminarayanan[1]*, Rakshit Shah[2], Sohail R. Daulat[3], Viashen Moodley[4], Yifei Yao[5], Srignana Lokesh Ezhil[1], Vadivelan Ramu[1], Puja Sengupta[1], Deepa Madathil[6]

1 Neuro-rehabilitation Lab, Department of Sensors and Biomedical Engineering, School of Electronics Engineering, Vellore Institute of Technology, Vellore, Tamil Nadu, India, 2 Department of Chemical and Biomedical Engineering, Cleveland State University, Cleveland, OH, United States of America, 3 University of Arizona College of Medicine–Tucson, Tucson, AZ, United States of America, 4 Arizona Center for Hand to Shoulder Surgery, Phoenix, AZ, United States of America, 5 Soft Tissue Biomechanics Laboratory, Med-X Research Institute, School of Biomedical Engineering, Shanghai Jiao Tong University, Shanghai, China, 6 Jindal Institute of Behavioural Sciences, O. P. Jindal Global University, Sonipat, Haryana, India

* KISHOR.LN@vit.ac.in

## Abstract

### Purpose

This study aimed to assess the usability of a virtual reality-assisted sensorimotor activation (VRSMA) apparatus for individual digit rehabilitation. The study had two main objectives: Firstly, to collect preliminary data on the expectations and preferences of patients with carpal tunnel syndrome (CTS) regarding virtual reality (VR) and an apparatus-assisted therapy for their affected digits. Secondly, to evaluate the usability of the VRSMA apparatus that was developed.

### Methods

The VRSMA system consists of an apparatus that provides sensory and motor stimulation via a vibratory motor and pressure sensor attached to a button, and a virtual reality-based visual cue provided by texts overlaid on top of a 3D model of a hand. The study involved 10 CTS patients who completed five blocks of VRSMA with their affected hand, with each block corresponding to the five digits. The patients were asked to complete a user expectations questionnaire before experiencing the VRSMA, and a user evaluation questionnaire after completing the VRSMA. Expectations for VRSMA were obtained from the questionnaire results using a House of Quality (HoQ) analysis.

### Results

In the survey for expectations, participants rated certain attributes as important for a rehabilitation device for CTS, with mean ratings above 4 for attributes such as ease of use, ease of understanding, motivation, and improvement of hand function based on clinical evidence. The level of immersion and an interesting rehabilitation regime received lower ratings, with mean ratings above 3.5. The survey evaluating VRSMA showed that the current prototype was overall satisfactory with a mean rating of 3.9 out of 5. Based on the HoQ matrix, the

(Grant number SRG/2021/000283) awarded to KL. The funders had no role in study design, data collection and analysis, decision to publish, or preparation of the manuscript.

**Competing interests:** The authors have declared that no competing interests exist.

highest priority for development of the VRSMA was to enhance device comfort and usage time. This was followed by the need to perform more clinical studies to provide evidence of the efficacy of the VRSMA. Other technical characteristics, such as VRSMA content and device reliability, had lower priority scores.

## Conclusion

The current study presents a potential for an individual digit sensorimotor rehabilitation device that is well-liked by CTS patients.

## Introduction

The goal of the current study was to evaluate the usability of a virtual reality-assisted sensorimotor activation (VRSMA) apparatus for individual digit rehabilitation. Specifically, the first objective was to gather initial data on the expectations and preferences of carpal tunnel syndrome (CTS) patients regarding virtual-reality (VR) and an apparatus assisted therapy for their affected digits. The second objective was to assess the usability of the VRSMA we developed.

CTS is associated with a range of abnormal peripheral motor functioning including motor weakness [1], heightened imprecision in force and motion during pinch [2], reduced range of motion [3], reduction in force coordination [4], and diminished dexterity of digits [2]. The compression of the median nerve inside the carpal tunnel has a direct impact on the motor supply to intrinsic hand muscles. With prolonged CTS, such nerve compression can ultimately lead to the loss or unconventional reorganization of the cortical representations in both the sensory and motor cortex. Such reorganization is similar to those observed in severe cases of nerve deafferentation resulting from conditions such as ischemic nerve block [5], spinal cord injury [6], prolonged limb immobilization [7], and cervical root avulsions [8]. Considering that the motor cortex mainly controls the transmission of descending corticospinal signals to the digits [9, 10], any maladaptive changes in the cortex could directly explain the loss of muscle or digit-related cortical representation. This, in turn, could significantly impact hand dexterity in individuals with chronic CTS.

Recent research suggests that CTS is not simply a peripheral neuropathy, but also involves cortical sensorimotor organization [11]. Compared to individuals who do not have CTS, those affected by the condition exhibit notable differences in their somatosensory representation [12, 13]. Additionally, entrapment neuropathies, like CTS, have been observed to lead to a decrease in intraepidermal nerve fiber density and changes in nodal structure and myelin that extend beyond the specific site of compression. Chronic pathological signaling in the afferent-efferent circuit of the CTS patient leads to changes in neural activity at multiple levels of the nervous system, which can impact somatotopic organization of individual digits. Poorly organized digit somatotopy is believed to be an indication of symptom severity and reduced tactile sensation, which may lead to digital discoordination and hand clumsiness [14]. Although the conventional understanding of CTS pathophysiology typically focuses on localized mechanisms at the site of the nerve compression, it is worth noting that as many as a quarter of patients continue to experience persistent or worsening symptoms even after undergoing surgical intervention [15].

The brain modulates sensorimotor neural pathways in response to sensory input and motor learning experience [16, 17], which has led to the development of rehabilitation

strategies that promote cortical plasticity to improve limb function. Peripheral sensory stimulation repetitively applied to the hand and foot has been shown to enhance tactile, haptic, discriminative, and sensorimotor performance in healthy individuals [18], patients with stroke [19], Parkinson's Disease [20], and spinal cord injury [21]. Repeated motor practice also promotes corticomotor output augmentation for centrally paretic hand and arm, inducing use-dependent brain plasticity to promote motor recovery [22]. It has been found that the most effective approach to neurorehabilitation for patients with sensorimotor deficits is through repetitive sensory stimulation of peripheral afferents combined with voluntary motor activation, i.e., closed-loop sensorimotor rehabilitation. This approach facilitates interconnections between the sensory and motor cortices, which has led to significant functional improvements in patients undergoing paired motor training with vibration stimulation [21, 23]. Closed-loop sensorimotor rehabilitation is particularly applicable to CTS patients who lack dexterous digit control, which can result in hand clumsiness and the dropping of handheld objects. To address this need, we have developed the VRSMA that engages cortical plasticity mechanisms via repetitive association of digit-specific motor responses with vibrotactile stimulation. The VRSMA achieves this by applying a vibrotactile stimulation to the digit pads as well as engaging the patients to perform voluntary force production using their digits while being directed by a virtual reality-based visual cue.

The VRSMA was evaluated for its usability by CTS patients using a House of Quality (HoQ) analysis. The use of the HoQ analysis as a design management tool allows for the design of products to be aligned with customers' desires right from the conception stage [24]. Prominent companies like Hewlett-Packard, AT&T, Ford, General Motors, and Toyota have used this process with proven results [24]. HoQ has also found practical applications, including the customization of VR based hand and arm rehabilitation games for stroke survivors [25] making it the best fit for evaluating VRSMA. The HoQ analysis helps identify the critical parameters of the device and translate these into design specifications that meet user needs and expectations [26]. The HoQ analysis process starts with identifying the needs and wants of CTS patients using the virtual reality assisted sensorimotor activation apparatus. These needs and wants are then translated into measurable parameters that the VRSMA should meet. For example, patients may need a device that is easy and comfortable to use. These needs may be translated into specific design specifications, such as a device with a clear user interface and foam paddings for the hand and wrist. The HoQ analysis also considers the relationship between these design specifications and how they affect each other. The current study utilized the HoQ approach to identify the key technical areas that require improvement in order to address user dissatisfaction and enhance user satisfaction. Adopting a user-centered approach was deemed necessary since many rehabilitation paradigms are developed without much input from the patients themselves. The preferences of the end-users are anticipated to play a significant role in the acceptance, motivation, and eventual outcomes of any rehabilitation program.

## Methods

### VRSMA

The VRSMA system consists of two components, namely the apparatus and the virtual reality-based visual cue. The apparatus (Fig 1) provided both sensory and motor stimulation via a button that has a vibratory motor and flexiforce sensor attached on top of the button. The sensory stimulation is a vibrotactile signal applied via small vibratory motors (Sunrobotics, Gujarat, India) on which the finger pads can be placed upon. The motor stimulation is provided by the momentary push button. The amount of pressure applied on the button can be monitored using a FlexiForce sensor (Norwood, MA, USA) placed on top of the button. Furthermore,

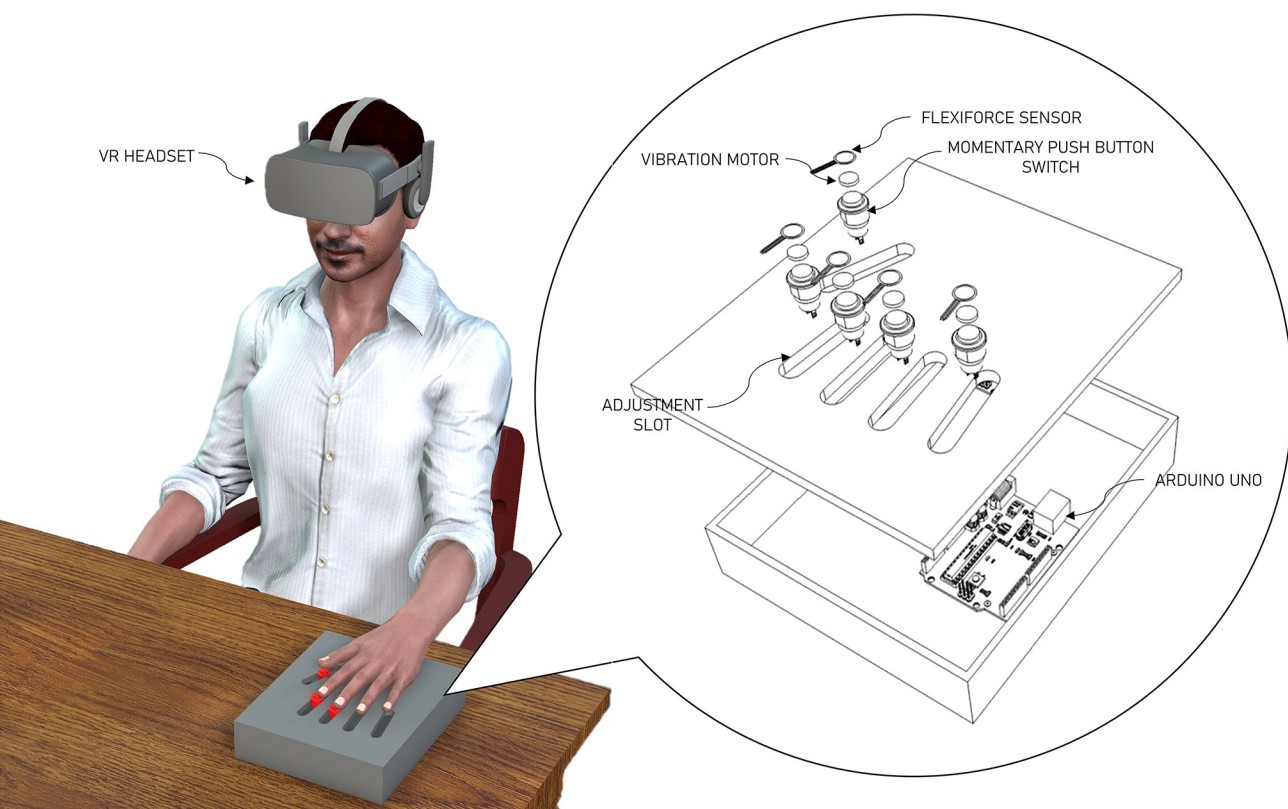

**Fig 1. VRSMA prototype.**

springs can be added or removed inside the button to control the amount of pressure required to completely press the button down. The buttons along with the vibratory motor and the pressure sensor are mounted in slots cut in a box. The users place their hand on top of the box while the button's position can be adjusted within the slot for each individual finger to align it with the finger pad. The system provides both sensory and motor stimulation that can be adjusted (vibration intensity and pressure required to push down the button) based on the study requirements and user's capabilities. The button, vibratory motor, and the pressure sensor are all controlled by an Arduino Uno (Arduino, Italy). The Arduino microcontroller is programmed to vibrate the motor for a short duration and also record the pressure sensor values and the time at which the button was pushed down and released.

The virtual reality-based visual cue was provided by texts overlaid on top of a 3D model of a hand that was modeled and animated in Blender software (Blender Foundation, Amsterdam, Netherlands). The 3D hand was animated to push down on the button using individual digits. The 3D hand was then exported to a virtual environment and gamified to perform the animation multiple times using Unity game engine (Unity Technologies, San Francisco, CA, USA). Users wore an Oculus Rift-S (Oculus VR, Menlo Park, CA, USA) virtual-reality headset that displayed the graphical scenario with the 3D hand performing the button pushing tasks in an immersive VR environment. At the start of a single trial a text was displayed on top of the 3D hand that informed the users on which digit they were going to receive the sensory stimulation and when. Following the sensory stimulation to a particular digit, another text cue directed the users to perform the motor task with the same finger while the animation assisted the user with the timing and pace of the voluntary motor activity. The VR provided the patients with an immersive environment to perform these tasks.

## Usability evaluation

**Subjects.**   Ten subjects (three females and seven males) with moderate to severe CTS symptoms based on the Boston Carpal Tunnel Syndrome Questionnaire (BCTQ) scores participated in the study. The protocol was approved by the Institutional Review Board. All subjects read and signed a written informed consent form before participating in the experiment.

**Procedure.**   The study aimed to obtain the expectations and preferences of CTS patients regarding the VRSMA. Prior to the experiment, personal interviews were conducted with five CTS patients who were given a detailed description of the device and its functionalities. During these interviews, the patients were asked to provide a list of their specific needs and requirements for the VRSMA. Based on these discussions, the identified subject requirements were translated into seven user attributes (Table 1). Subsequently, a user expectations questionnaire was developed that consisted of these attributes. The participants could rate the level of importance of each attribute on a five-point Likert scale (ranging from 1- not important, to 5- very important) in terms of enhancing the device's performance.

To evaluate the usability of the VRSMA, each of the ten CTS patients involved in the study was asked to visit the lab. After obtaining their consent, the patients were briefed on the details of the device, and the user expectations questionnaire was administered to them. They were then asked to experience the VRSMA while they engaged in sensory stimulation and voluntary motor activity for two hours.

Each patient experienced the VRSMA with their affected hand. An instruction manual was presented to each patient on how to wear the VR headset and place their affected hand over the apparatus. Additionally, the manual included initial findings that suggested immediate improvements in reaction time to a vibrotactile stimulus in healthy participants [27]. Assistance was provided upon request. Each CTS patient completed five blocks of VRSMA, corresponding to the five digits. Each block consisted of five sessions, and each session included 10 trials of VRSMA with adequate rest provided between sessions. At the start of each trial, a text reminded the subject which digit was being tested, and a brief vibrotactile stimulation lasting one second was administered to the corresponding finger pad. The trial then proceeded with a 3-second waiting period instructing the participant to be ready, followed by an animation showing the digit pressing down on a button, holding the position for 2 seconds, and then releasing the button to return to the initial position. This was followed by a 3-second rest. Participants were asked to observe the task animation and perform the motor task at the same pace as the 3D model.

Following this experience, the patients were given the user evaluation questionnaire, which contained the same attributes as the user expectations questionnaire, and were asked to rate the device's performance on a five-point Likert scale. Overall, the study utilized a user-centered approach to identify the key requirements and attributes that are important for the VRSMA's

**Table 1.  User attributes.**

| User attributes |
| --- |
| Easy to Understand |
| Easy to Use |
| Interesting |
| Motivating |
| Level of Immersion |
| User Improvement |
| Clinical Evidence |

success. This approach included personal interviews, the development of questionnaires, and the evaluation of the device's usability through patient feedback.

**HoQ analysis.** The information gathered from the subjects through the questionnaires was used to construct the House of Quality. The user expectations block of the HoQ consisted of the attributes identified through the user expectations questionnaire and their respective weights. To obtain the weight for each attribute, the rating for each attribute was averaged over all the subjects. This weight indicated the relative importance of each attribute based on the feedback received from the subjects. The next step involved matching the technical characteristics of the VRSMA with the user attributes identified in the user expectations block. Eight technical characteristics were identified based on an engineering point of view. A relationship matrix was created to establish the relationship between the technical characteristics and the user attributes. The strength of the relationship between each user attribute and each technical characteristic was determined and rated as small, medium, or strong, with corresponding numeric values of 1, 3, or 9. An empty cell indicated no relationship.

Using the weight of each user attribute and its relationship score with each technical characteristic, the importance weight of each technical characteristic was calculated. For instance, the importance weight of VRSMA ease of use was computed as 44.2, which was obtained by multiplying the weight for Easy to use (4.5) by the relationship score for VRSMA usage length (9) and adding it to the weight for Level of immersion (3.7) multiplied by the relationship score for VRSMA ease of use (1). Once the importance weight of each technical characteristic was calculated, the percentage of importance of each technical characteristic was determined by normalizing the importance weight to the sum of all importance weights across all technical characteristics. The user evaluation rating for each user expectation attribute was calculated by averaging the score for each attribute received in the user evaluation questionnaire over all the subjects.

To obtain the priority weight of each technical characteristic, the weight of each user attribute, its relationship scores with the technical characteristic, and the user evaluation rating for that attribute were considered. For instance, the priority weight of VRSMA ease of use was computed as 62.4, which was obtained by multiplying the weight for Easy to use (4.5) by the relationship score for VRSMA ease of use (9) and the user rating (5–3.5) and adding it to the weight for Level of immersion (3.7) multiplied by the relationship score for VRSMA ease of use (1) and the user rating (5–4). Finally, the priority percentage was estimated the same way as importance percentage to determine the priority of each technical characteristic in improving the current VRSMA.

## Results

### Expectations for VRSMA

Fig 2A presents the findings of the user expectation questionnaire used in the study. The results indicate that the subjects considered certain attributes as important for a rehabilitation device for CTS. Specifically, the participants identified that it is very important (with mean rating > 4) for the device to be easy to use, easy to understand, motivating, and helpful in improving hand function based on clinical evidence. The criteria that did not receive much importance (with mean rating > 3.5) were the level of immersion and an interesting rehabilitation regime.

### Evaluation of VRSMA

A detailed analysis of the user evaluation of VRSMA was conducted by examining the ratings provided by the subjects after they experienced the device (as shown in Fig 2B). Overall, the device performed quite well in the user ratings with most aspects getting a score of 4 or above.

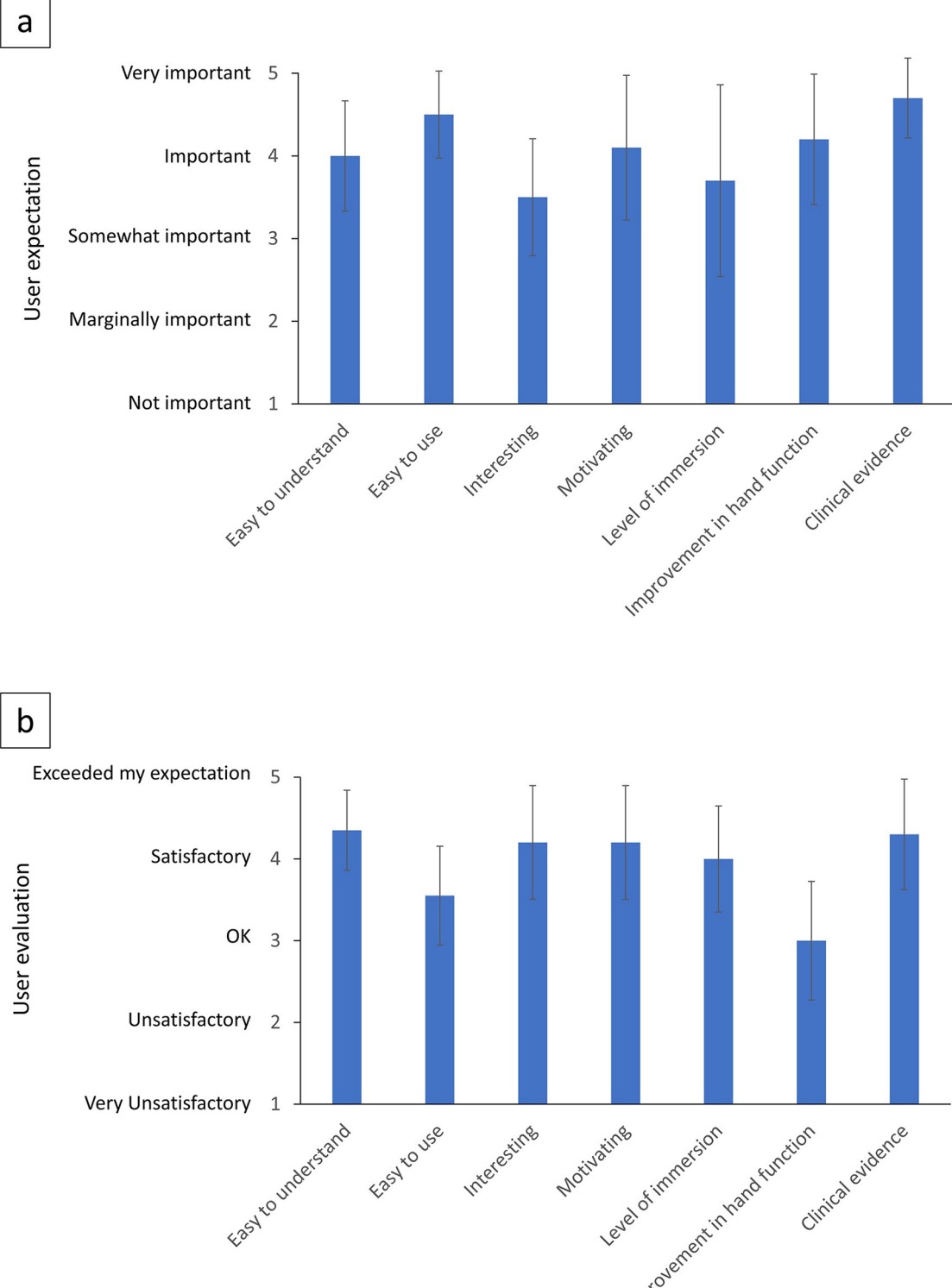

**Fig 2.** a) Importance of each of 7 criteria for a digit rehabilitation apparatus. b) User satisfaction with the VRSMA prototype for individual criteria. Mean ± standard errors from 10 CTS patients' ratings are shown.

According to the ratings, the user improvement received a mean low rating of 3, ease of use received a score of 3.5, while level of immersion got a score of 4. The device was rated highly for its ease of understanding, which matched the subjects' initial expectations.

## Priority development needs

The data gathered from the user expectations and evaluation was utilized to construct a HoQ matrix (Fig 3A), which is a tool used to identify the technical characteristics of the VRSMA that require the highest priority for development (Fig 3B). The results of the HoQ analysis revealed that the most pressing development need was to enhance the device comfort and usage time, followed by the need to perform more clinical studies to provide evidence of the efficacy of the VRSMA. The HoQ matrix also showed that other technical characteristics, such as VRSMA content and device reliability were important but had a lower priority than the comfort and clinical studies.

## Discussion

Multiple-digit sensorimotor choice reaction times are a cutting-edge and effective way to assess central processing deficiency in CTS. Due in part to the numerous practical uses for auditory (such as running events) and visual (such as obstacle avoidance) responses, reaction time is frequently researched using auditory or visual stimuli. Research on sensorimotor reaction time, specifically response to a tactile stimulus administered peripherally (i.e. at the fingertip), is currently limited. Tactile stimulation and tactile imagery given to the tips of the digits during multi-digit sensorimotor choice reaction time has shown an increase in reaction time [27] and motor learning [28] in healthy volunteers, however, such information on CTS patients is limited. Given the sensory and motor pathways involved in this nerve, tactile stimulation might be a crucial first step in improving sensorimotor reaction time in CTS.

It has also been demonstrated that regular motor practice helps improve functional and corticomotor output in centrally paretic hands and arms. To promote use-dependent brain plasticity and motor recovery in individuals with movement disorders, constraint-induced movement therapy, for instance, enhances movement practice [29]. Motor imagery has been shown to reduce corticomotor excitability and enhance motor function [30]. By promoting connections between the sensory and motor cortices, recurrent sensory stimulation of peripheral afferents in conjunction with voluntary motor activation may be the most beneficial method of neurorehabilitation for patients who already have motor deficits [31, 32].

The VRSMA developed in this study engages cortical plasticity mechanisms via repetitive association of digit-specific motor responses with vibrotactile stimulation. The VRSMA achieves this by applying a vibrotactile stimulation to the digit pads and engaging the patients to perform voluntary force production using their digits while being directed by a virtual reality-based visual cue. VR-based physical rehabilitation in patients with SCI improved motor performance, compared to standard physical rehabilitation [33] and improved motor imagery performance [30]. Additionally, MRI studies have demonstrated cortical plastic changes after VR-based physical rehabilitation in SCI subjects [34]. The use of virtual reality-based rehabilitation paradigm for CTS has not been attempted before to our knowledge. Hence, it is important to evaluate the usability of such a paradigm with CTS patients. The expectations and user rating from the patients will help shape the prototype design.

In terms of expectations for the VRSMA, CTS patients regarded ease of use and control, understanding, and effectiveness in improving hand function as the most important criteria, while the need for the VRSMA to be interesting was considered the least important followed by the level of immersion the VR-based graphical cues needed to induce (Fig 2A). The CTS

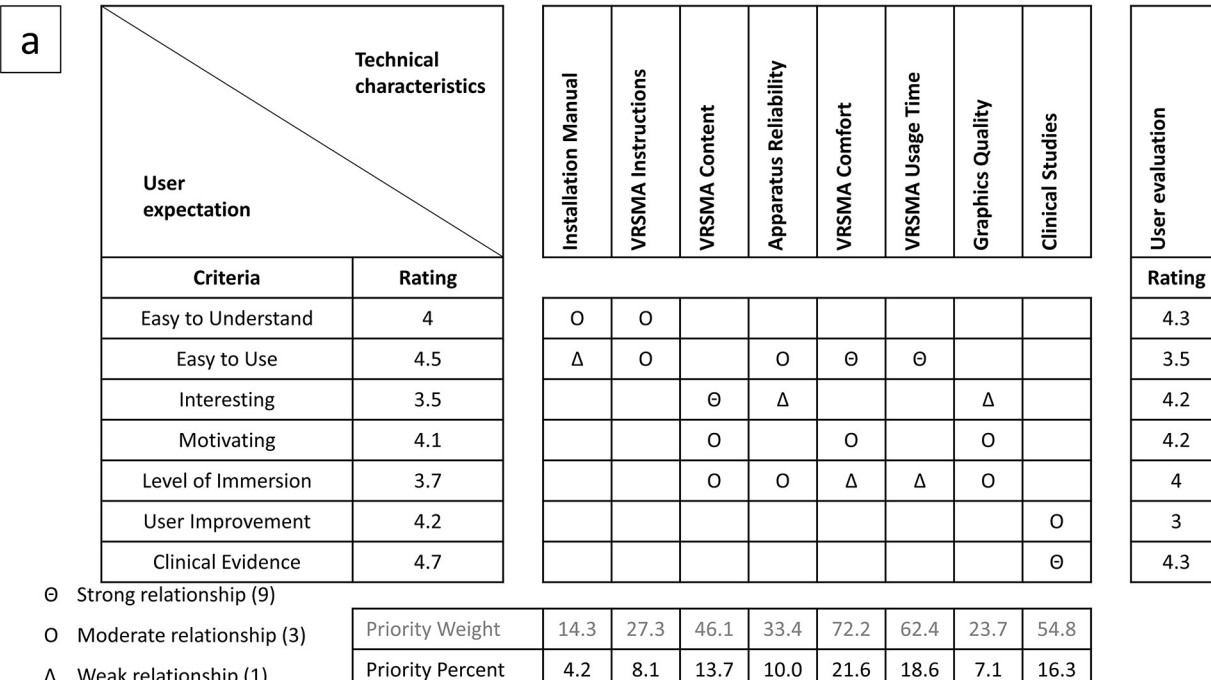

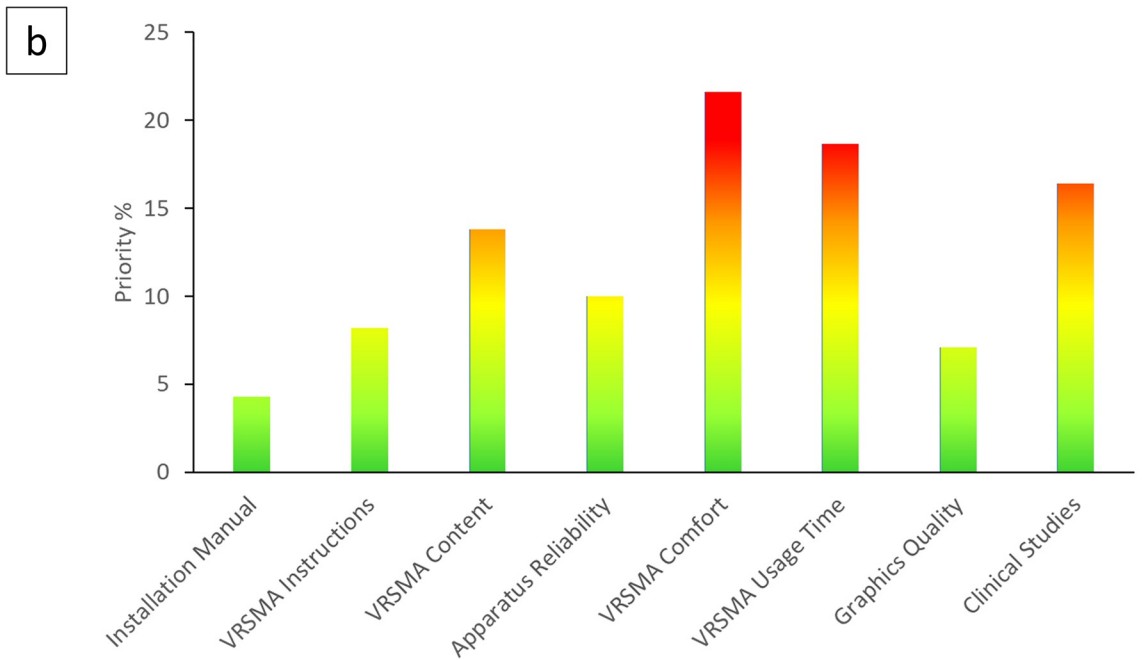

**Fig 3.** a) House of Quality matrix. b) priority percent showing the relative priority of improvement needs.

patients who participated in the study gave a mean rating of 3.9 out of 5 for the VRSMA prototype, indicating that the device was well received and was close to satisfactory. The ease of understanding especially received a very high rating of 4.3. Followed by the clinical evidence from the device we showed for the feasibility of the rehabilitation regime the device offered in improving the reaction time in healthy individuals [27]. However, the ease of use and user improvement from the usage of the device were aspects the patients scored the least with a mean rating of less than 4.

Future development efforts should focus on the top priority improvement needs identified in the HoQ matrix, as shown in Fig 3A. These include improving the comfort of the device, the length of time patients would be able to use the device comfortably, and the clinical knowledge. Ease of use was one of the least scoring attributes of the VRSMA. The current prototype of the VRSMA apparatus is a wooden box with slots cut in to adjust for each person's hand size. Additionally, an off-the-shelf virtual-reality headset was used in the study that was connected to a computer to display the graphical cues. A post survey discussion with the patients revealed that the VR headset was heavy and not comfortable to wear. Although patients were appreciative of the graphical cue and the level of immersion the VR offered, they still did not feel comfortable wearing the headset for extended periods of time. The current prototype was designed to be portable and to be used at home by the patients without the need to visit a clinic. However, based on the HoQ analysis, the future prototype will aim for a less bulky design for the apparatus and an untethered VR glass that is both lightweight and uses a smartphone to display the graphical cues. A VR glass only costs around $20 compared to the expensive VR headset and a high-end computer required to run the headset. A wearable version of the VRSMA device could possibly be incorporated in a glove with a vibrator attached to the finger pads of the glove with a button underneath it that can be pressed against any hard surface. Such a wearable version of the device would greatly reduce the bulkiness of the device and make it more portable. While a lightweight, untethered VR glass would add to the portability of the device.

User improvement was the least scoring criterion, although it was rated as "OK" with a mean rating of 3. The patients experienced VRSMA for two hours with one hour each for the sensory stimulation and motor activity. In spite of the fact that the discomfort of wearing the VR headset for such a long period might have influenced the ratings for the user improvement, only 3 out of the 10 patients reported that they felt no change after using the device. Future studies should evaluate the required number of sessions and length of rehabilitation required for patients to see considerable change in their symptoms. Improved clinical knowledge for specific individuals' characteristics was identified as a top priority improvement need.

## Conclusion

The current study demonstrated the feasibility of a virtual reality-assisted sensorimotor activation apparatus for individual digit rehabilitation. CTS patients expressed a willingness to try this device and the current prototype was well-received by the patients, receiving a mean rating of 3.9 out of 5 (close to satisfactory). The House of Quality analysis indicated that the prototype's priority improvement needs include improving comfort in using the device, length of time the device could be used, and enhancing clinical knowledge on long-term effectiveness. In conclusion, this study presents a promising option for a novel digit rehabilitation device that is well-liked by CTS patients.

## Acknowledgments

The authors would like to thank Med-Direct LLC for providing help with the HoQ Analysis.

## Author Contributions

**Conceptualization:** Kishor Lakshminarayanan, Yifei Yao.

**Data curation:** Srignana Lokesh Ezhil, Vadivelan Ramu, Puja Sengupta.

**Formal analysis:** Kishor Lakshminarayanan, Rakshit Shah, Sohail R. Daulat, Viashen Moodley, Deepa Madathil.

**Funding acquisition:** Kishor Lakshminarayanan.

**Writing – original draft:** Kishor Lakshminarayanan.

**Writing – review & editing:** Rakshit Shah, Sohail R. Daulat, Viashen Moodley, Yifei Yao, Srignana Lokesh Ezhil, Vadivelan Ramu, Puja Sengupta, Deepa Madathil.

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
