## [Decision Letter · Decision Letter 0]

29 Aug 2023

PONE-D-23-09552

Feasibility and Usability of a Virtual-Reality-based Sensorimotor Activation Apparatus for Carpal Tunnel Syndrome Patients

PLOS ONE

Dear Dr. Lakshminarayanan,

Thank you for submitting your manuscript to PLOS ONE. After careful consideration, we feel that it has merit but does not fully meet PLOS ONE’s publication criteria as it currently stands. Therefore, we invite you to submit a revised version of the manuscript that addresses the points raised during the review process.

In the light of reviewers' comments and feedback.

We look forward to receiving your revised manuscript.

Kind regards,

Umer Asgher, PhD

Academic Editor

PLOS ONE

Journal Requirements:

Reviewers' comments:

Reviewer's Responses to Questions

**Comments to the Author**

1. Is the manuscript technically sound, and do the data support the conclusions?

Reviewer #1: Yes

Reviewer #2: Yes

Reviewer #3: Yes

2. Has the statistical analysis been performed appropriately and rigorously? 

Reviewer #1: No

Reviewer #2: Yes

Reviewer #3: Yes

3. Have the authors made all data underlying the findings in their manuscript fully available?

Reviewer #1: Yes

Reviewer #2: Yes

Reviewer #3: Yes

4. Is the manuscript presented in an intelligible fashion and written in standard English?

Reviewer #1: Yes

Reviewer #2: Yes

Reviewer #3: Yes

5. Review Comments to the Author

Reviewer #1: This feasibility and usability study reports preliminary results from ten CT patients who tried a VR-based sensorimotor activation apparatus for hand function rehab.

Authors did not cite enough literatures of VR or computer-assisted rehabilitation among CT in the Introduction nor in the Discussion section. Thus, it is not clear whether this study will advance the field or not.

Given the small sample size and preliminary nature of the results, PLOS One may not be a good choice for publishing it. Authors are encouraged to seek other alternative journals.

Reviewer #2: The study evaluates the usability of a virtual reality-assisted sensorimotor activation (VRSMA) apparatus for rehabilitating individual digits in carpal tunnel syndrome (CTS) patients. It aims to gather patient preferences for VR-based therapy and assesses how the VRSMA engages cortical plasticity mechanisms. The VRSMA combines vibrotactile stimulation and virtual reality cues to enhance digit control. The study uses a House of Quality (HoQ) analysis to ensure the device meets patient needs. The research seeks to improve limb function and quality of life for CTS patients with impaired digit control. The paper is well-written. I've a few minor suggestions which I've listed below.

1. The introduction mentions that the VRSMA addresses the need for patients with dexterous digit control issues, but it could expand on why this is an important area of focus in CTS rehabilitation. Emphasize the potential impact and benefits of the VRSMA on patients' quality of life and functional improvements.

2. While the use of House of Quality (HoQ) analysis is mentioned as a design management tool, the passage should include a clearer justification for choosing this method over other usability assessment techniques. Explain how HoQ analysis specifically aligns with the study's objectives and the advantages it offers.

3. The usability evaluation included CTS patients with various levels of severity but the device seems more useful to the mild to moderate CTS population since severe cases would most probably benefit only from surgical intervention. Why include severe cases in the current study?

4. The discussion mentions that user improvement was rated lower and might have been influenced by the discomfort of wearing the VR headset. To address this concern, the study could explore ways to mitigate discomfort during extended usage, as mentioned in the wearable version suggestion.

Reviewer #3: In this study, authors have used a virtual reality based sensory motor activation apparatus for digit rehabilitation primarily focused on Carpal Tunel Syndrome patients. The study seems to be thorough and manuscript is well articulated and well written. However, there are some minor comments as below:

1. In the introduction section, please elaborate the principle and mechanism of VRSMA. A schematic of the mechanism of how the whole set up is acting on the brain and stimulating the sensory motor activation would be helpful.

2. Limited literature cited in the introduction section and whole manuscript in general. Also most of the literature cited are older. Including latest works is suggested.

3. The authors evaluated the device's usability and feasibility including CTS patients of both genders with moderate to severe pain in the affected hand. As it involves the movement of the digits, the majority of CTS patients experience severe pain to the point where they cannot even move their fingers. So, the authors could have provided a severity scale so that the device could be beneficial for those who require rehabilitation or who can benefit from it at an early recovering stage.

4. Did authors consider assessing the difference in outcome between male and female? If yes, what is the result and if not why?

5.In the discussion section, the authors attempted to describe the design of the device, but it is unclear whether the CTS patient can use the device at home or only under the supervision of a medical practitioner? Also are the device and its parts portable enough for use at home? Please calrify

6. How was the outcome of this device validated? Please clarify. Even though this approach seems promising, validating the usability and outcome with the existing standard of care would add more merit to this work.

6. PLOS authors have the option to publish the peer review history of their article (what does this mean?). If published, this will include your full peer review and any attached files.

Reviewer #1: No

Reviewer #2: **Yes: **Gautam Mahajan

Reviewer #3: **Yes: **Shataakshi Dahal

---

## [Author Response · Author response to Decision Letter 0]

1 Sep 2023

Reviewer #1: This feasibility and usability study reports preliminary results from ten CT patients who tried a VR-based sensorimotor activation apparatus for hand function rehab.

Authors did not cite enough literatures of VR or computer-assisted rehabilitation among CT in the Introduction nor in the Discussion section. Thus, it is not clear whether this study will advance the field or not.

Response 1: We thank the reviewer for suggesting comments to improve our work. We have now added citations about VR-based rehab in patient populations such as spinal cord injured patients. however, to our knowledge, there is no existing rehab regime using VR with CT patients. We have also added citations on the effect of sensorimotor training in CTS patients.

Given the small sample size and preliminary nature of the results, PLOS One may not be a good choice for publishing it. Authors are encouraged to seek other alternative journals.

Response 2: We believe the sample size of 10 CTS patients, though small, is enough for a usability evaluation based on previous studies that were published in PLOSOne too among other equivalent journals. We are including few of those studies here for the kind perusal of the reviewer. We implore the reviewer to reconsider their opinion on the study being unfit for the journal.

• Zhou Z, Li J, Wang H, Luan Z, Li Y, Peng X. Upper limb rehabilitation system based on virtual reality for breast cancer patients: Development and usability study. PloS one. 2021 Dec 15;16(12):e0261220.

• Seo NJ, Kumar JA, Hur P, Crocher V, Motawar B, Lakshminarayanan K. Usability evaluation of low-cost virtual reality hand and arm rehabilitation games. Journal of rehabilitation research and development. 2016;53(3):321.

• Lakshminarayanan K, Wang F, Webster JG, Seo NJ. Feasibility and usability of a wearable orthotic for stroke survivors with hand impairment. Disability and Rehabilitation: Assistive Technology. 2017 Feb 17;12(2):175-83.

 

Reviewer #2: The study evaluates the usability of a virtual reality-assisted sensorimotor activation (VRSMA) apparatus for rehabilitating individual digits in carpal tunnel syndrome (CTS) patients. It aims to gather patient preferences for VR-based therapy and assesses how the VRSMA engages cortical plasticity mechanisms. The VRSMA combines vibrotactile stimulation and virtual reality cues to enhance digit control. The study uses a House of Quality (HoQ) analysis to ensure the device meets patient needs. The research seeks to improve limb function and quality of life for CTS patients with impaired digit control. The paper is well-written. I've a few minor suggestions which I've listed below.

1. The introduction mentions that the VRSMA addresses the need for patients with dexterous digit control issues, but it could expand on why this is an important area of focus in CTS rehabilitation. Emphasize the potential impact and benefits of the VRSMA on patients' quality of life and functional improvements.

Response 3: We thank the reviewer for the insightful comments. We have addressed this comment by expanding on the importance of sensorimotor rehabilitation in CTS patients in the introduction. The new information are highlighted in yellow.

2. While the use of House of Quality (HoQ) analysis is mentioned as a design management tool, the passage should include a clearer justification for choosing this method over other usability assessment techniques. Explain how HoQ analysis specifically aligns with the study's objectives and the advantages it offers.

Response 4: Indeed, our decision to use HoQ is based on previous studies that have used HoQ for evaluating VR-based rehab games, making it a good fit for our purpose. Furthermore, HoQ aligns user expectations with engineering needs which makes it unique in prototype development. We have included the information in the introduction.

3. The usability evaluation included CTS patients with various levels of severity but the device seems more useful to the mild to moderate CTS population since severe cases would most probably benefit only from surgical intervention. Why include severe cases in the current study?

Response 5: The reviewer is correct in noting that severe cases of CTS are mostly recommended surgical intervention. Our reasoning for including them in the study is two-fold:

a) Post-surgical rehabilitation largely includes rehab exercises in CTS patients and as such the VRSMA might be used as part of post-surgery rehab.

b) Furthermore, non-surgical treatment is now largely developed and recommended for sever CTS patients to manage their symptoms, including rehabilitation exercises. Hence there is a possibility for such patients to benefit (even if not in a profound way) from our VRSMA.

We included severe CTS patients too to first get a comprehensive understanding of user expectations, although not all severity levels might benefit completely from the current device. 

4. The discussion mentions that user improvement was rated lower and might have been influenced by the discomfort of wearing the VR headset. To address this concern, the study could explore ways to mitigate discomfort during extended usage, as mentioned in the wearable version suggestion.

Response 6: Indeed, we have included addressing discomfort from wearing VR headset as a priority. A possible solution is using VR glasses that are cheap and lightweight as an alternative to VR headsets. We will explore the alternatives in the next prototype.

 

Reviewer #3: In this study, authors have used a virtual reality based sensory motor activation apparatus for digit rehabilitation primarily focused on Carpal Tunel Syndrome patients. The study seems to be thorough and manuscript is well articulated and well written. However, there are some minor comments as below:

1. In the introduction section, please elaborate the principle and mechanism of VRSMA. A schematic of the mechanism of how the whole set up is acting on the brain and stimulating the sensory motor activation would be helpful.

Response 7: We have added information in the introduction on the principle and mechanism through which VRSMA could improve functionality in CTS patients. The newly added information is highlighted in yellow.

2. Limited literature cited in the introduction section and whole manuscript in general. Also most of the literature cited are older. Including latest works is suggested.

Response 8: More citations including latest works have been cited now.

3. The authors evaluated the device's usability and feasibility including CTS patients of both genders with moderate to severe pain in the affected hand. As it involves the movement of the digits, the majority of CTS patients experience severe pain to the point where they cannot even move their fingers. So, the authors could have provided a severity scale so that the device could be beneficial for those who require rehabilitation or who can benefit from it at an early recovering stage.

Response 9: We thank the reviewer for the suggestion. We are currently working on including an interactive Boston Carpal Tunnel Questionnaire (BCTQ) that will be displayed in the VR headset and based on the patients severity the number of exercises and their repetitions will be set automatically.

4. Did authors consider assessing the difference in outcome between male and female? If yes, what is the result and if not why?

Response 10: Although the reviewer is correct in noting there is a sex-based prevalence in CTS, such prevalence is mostly seen in the occurrence of CTS only but not in the prognosis. The current study is to only evaluate the usability of a CTS rehab device and as such we believe there is no merit in evaluating any sex-based differences. Future studies evaluating the effectiveness of the rehab regime in improving sensorimotor function in CTS patients can evaluate any sex-based differences.

5.In the discussion section, the authors attempted to describe the design of the device, but it is unclear whether the CTS patient can use the device at home or only under the supervision of a medical practitioner? Also are the device and its parts portable enough for use at home? Please clarify

Response 11: The device is portable and can be used at home. We have now included the information in the discussion section.

6. How was the outcome of this device validated? Please clarify. Even though this approach seems promising, validating the usability and outcome with the existing standard of care would add more merit to this work.

Response 12: The current study evaluated the feasibility and usability of a rehab device and as such there are no outcomes about the performance of the device as a rehabilitation tool itself but rather how is the usability of the device. The device’s performance has been evaluated in another study in improving sensorimotor function.

• Lakshminarayanan K, Ramu V, Rajendran J, Chandrasekaran KP, Shah R, Daulat SR, Moodley V, Madathil D. The effect of tactile imagery training on reaction time in healthy participants. Brain Sciences. 2023 Feb 14;13(2):321.

In the current study we only evaluated the usability of the device and method itself. A comparative study of the device’s performance with standard of care will be performed in the future.

---

## [Decision Letter · Decision Letter 1]

21 Sep 2023

Feasibility and Usability of a Virtual-Reality-based Sensorimotor Activation Apparatus for Carpal Tunnel Syndrome Patients

PONE-D-23-09552R1

Dear Dr. Lakshminarayanan,

We’re pleased to inform you that your manuscript has been judged scientifically suitable for publication and will be formally accepted for publication once it meets all outstanding technical requirements.

Kind regards,

Umer Asgher, PhD

Academic Editor

PLOS ONE

Additional Editor Comments (optional):

Reviewers' comments:

Reviewer's Responses to Questions

**Comments to the Author**

1. If the authors have adequately addressed your comments raised in a previous round of review and you feel that this manuscript is now acceptable for publication, you may indicate that here to bypass the “Comments to the Author” section, enter your conflict of interest statement in the “Confidential to Editor” section, and submit your "Accept" recommendation.

Reviewer #2: All comments have been addressed

Reviewer #3: All comments have been addressed

2. Is the manuscript technically sound, and do the data support the conclusions?

Reviewer #2: Yes

Reviewer #3: Yes

3. Has the statistical analysis been performed appropriately and rigorously? 

Reviewer #2: Yes

Reviewer #3: Yes

4. Have the authors made all data underlying the findings in their manuscript fully available?

Reviewer #2: Yes

Reviewer #3: Yes

5. Is the manuscript presented in an intelligible fashion and written in standard English?

Reviewer #2: Yes

Reviewer #3: Yes

6. Review Comments to the Author

Reviewer #2: (No Response)

Reviewer #3: I thank the authors for addressing all the comments. However, there is a small suggestion.

1. Please mention the line number in the response in addition to mentioning it was highlighted in yellow for easy navigation. The yellow highlight did not appear in the copy received for revision.

2. For the updated reference, please mention in the response page which reference numbers were the updated ones.

7. PLOS authors have the option to publish the peer review history of their article (what does this mean?). If published, this will include your full peer review and any attached files.

Reviewer #2: **Yes: **Gautam Mahajan

Reviewer #3: No

---

## [Editor Report · Acceptance letter]

3 Oct 2023

PONE-D-23-09552R1 

Feasibility and Usability of a Virtual-Reality-based Sensorimotor Activation Apparatus for Carpal Tunnel Syndrome Patients 

Dear Dr. Lakshminarayanan:

I'm pleased to inform you that your manuscript has been deemed suitable for publication in PLOS ONE. Congratulations! Your manuscript is now with our production department. 

Kind regards, 

on behalf of

Dr. Umer Asgher 

Academic Editor

PLOS ONE